# Copper-catalyzed *Z*-selective synthesis of acrylamides and polyacrylamides via alkylidene ketenimines

Xuelun Duan[1,2], Nan Zheng [ID][2,3] ✉, Ming Li[2,3], Gongbo Liu[1,2], Xinhao Sun[1,2], Qiming Wu[1,2] & Wangze Song [ID][1,2] ✉

It remains very important to discover and study new fundamental intermediates consisting of carbon and nitrogen as the abundant elements of organic molecules. The unique alkylidene ketenimine could be formed in situ under mild conditions by an unexpected copper-catalyzed three-component reaction of alkyne, azide and water involving a successive cycloaddition, $N_2$ extrusion and carbene-assisted rearrangement. Only *Z*-α,β-unsaturated amides instead of *E*-α,β-unsaturated amides or triazoles were acquired from alkylidene ketenimines with excellent selectivities and stereospecificities. In addition, a series of "approximate" alternating copolymers (poly (triazole-*alt*-*Z*-acrylamides)) with high $M_n$s and yields were efficiently afforded by multicomponent polymerization through a very simple operation basing on this multicomponent reaction.

Alkylidene ketenimines, as the unique cumulated ketenimines with general formula "$R_2C=C=C=NR$", are rarely reported than the well-known ketenimine intermediates, let alone further transformations of alkylidene ketenimines[1,2]. Compared to the unusual alkylidene ketenimines, ketenimine intermediates could be widely converted to various useful compounds and materials[3,4]. Chang and Fokin et al. first discovered ketenimines could be quickly formed in situ by copper-catalyzed electron-withdrawing azide-alkyne cycloadditions[5,6]. We and others have also been engaged to convert ketenimines using various nucleophiles and electrophiles which rapidly promoted the burgeoning growth of multicomponent reactions (MCR) in organic chemistry[7–13] and multicomponent polymerizations (MCP) in polymer science[14–21] involving ketenimine intermediates (Fig. 1a). However, the strong electron-withdrawing groups such as sulfonyl, phosphoryl and acyl groups were inevitable for azides to afford ketenimines by extruding $N_2$, which profoundly restricted the versatility of products[3–23]. Therefore, it is still highly desired to achieve the release of $N_2$ from more general azide via a facile reactive intermediate under mild conditions. Alkylidene ketenimines, as highly reactive intermediates, are traditionally considered to be formed under very

harsh conditions like flash vacuum pyrolysis[1,2]. As far as we know, only two examples were previously reported involving alkylidene ketenimine. Sandmeier et al. disclosed a [2 + 2] cycloaddition reaction despite using only one example of alkylidene ketenimine in 1975[24]. Until recently, Goldup et al. designed a very special tandem active template to afford *E*-acrylamide rotaxanes by complicated supramolecular technology. Unfortunately, no acrylamide was generated without using active template for non-interlocked products due that this process was dictated by mechanical bond. Even worse for the *E*-acrylamide rotaxanes products, low selectivities (acrylamide-triazole) and narrow substrate scope with bulky groups (5 bulky acrylamides) severely limited further applications of this supramolecular method to access more diverse acrylamides (Fig. 1b)[25]. Here, we show an unexpected copper-catalyzed three-component reaction of alkyne, azide and water by a successive domino process including cycloaddition, $N_2$ extrusion and carbene-assisted rearrangement with the formation of alkylidene ketenimine intermediate in situ (Fig. 1c). More interestingly, we can selectively control the types of products as *Z*-acrylamides or triazoles by electronic effect of substrates.

[1]State Key Laboratory of Fine Chemicals, School of Chemical Engineering, Dalian University of Technology, 116024 Dalian, China. [2]Department of Pharmaceutical Science, School of Chemical Engineering, Dalian University of Technology, 116024 Dalian, China. [3]Department of Polymer Science & Materials, School of Chemical Engineering, Dalian University of Technology, 116024 Dalian, China. ✉e-mail: nzheng@dlut.edu.cn; wzsong@dlut.edu.cn

**Fig. 1 | The formation and transformation for alkylidene ketenimine. a** Well-developed ketenimine chemistry. **b** Rarely reported alkylidene ketenimine. **c** Our approach for selective synthesis of *Z*-acrylamides involving alkylidene ketenimine.

$\alpha,\beta$-Unsaturated amide is one of the most useful and fundamental building blocks for organic synthesis, drug discovery and functional materials[26–28]. Natural products containing acrylamide moieties like cinnamides exhibit a variety of bioactivities, including anti-inflammatory, anti-microbial, and anti-tumor properties[29–32]. Compared with *E*-isomer, the *Z*-selective acrylamides could be widely used as unique building blocks for synthetic transformations and drug delivery, or as the important structures for bioactive molecules such as motualevic acid B and basiliskamide A[33–37]. However, compared to the well-defined preparation of *E*-acrylamide, highly stereospecific synthesis of *Z*-isomer remains a formidable challenge due to the large thermodynamic gap[37]. *Z*-$\alpha,\beta$-unsaturated amides could be obtained by very limited approaches such as Wittig reaction and its variants, photoisomerization, metathesis and so on regardless of poor *Z/E* selectivities and narrow substrate range sometimes[37–40].

Herein, we address above issues to access *Z*-acrylamides and poly-*Z*-acrylamides libraries by highly selective transformations of alkylidene ketenimines under mild conditions. Only *Z*-$\alpha,\beta$-unsaturated amides instead of *E*-$\alpha,\beta$-unsaturated amides are afforded from alkylidene ketenimines with excellent stereospecificities (*Z:E* > 19:1) and broad substrate scope (36 examples) by both MCR and MCP (Fig. 1c). We believe this strategy could inspire more useful synthetic

methodologies developed in organic chemistry, polymer science and other areas with thriving alkylidene ketenimine chemistry in future.

## Results and discussion
### Optimization of reaction conditions

For the preparation of *Z*-acrylamide (**3**), *tert*-butyl (1-phenylprop-2-yn-1-yl) carbonate (**1a**) and benzyl azide (**2a**) were initially chosen as the model substrates for optimizing the reaction conditions in Table 1. A series of commercially available coppers(I) were investigated. Cu(MeCN)$_4$BF$_4$ could catalyze this transformation with higher yield than CuOTf (Table 1, entries 1–2). To our delight, CuI and CuBr were demonstrated more efficient than Cu(MeCN)$_4$BF$_4$ and CuOTf for this reaction (Table 1, entries 3–4). CuBr was identified as the best catalyst for obtaining **3a** in good yield and excellent selectivity (*Z:E* > 19:1) (Table 1, entry 4). The yield could not be further improved by increasing the catalyst loading to 20 mol% or decreasing to 5 mol% (Table 1, entries 5–6). Various bases were next screened (Table 1, entries 7–11). The reaction failed to occur in the absence of base, indicating that base was required for this transformation (Table 1, entry 7). Triethylamine (TEA) could better promote this process compared with 4-dimethylaminopyridine (DMAP) and 1,8-diazabicyclo[5.4.0]undec-7-ene (DBU) (Table 1, entries 8–10). K$_2$CO$_3$ as

**Table 1 | Optimization of Reaction Conditions[a]**

| Entry | Catalyst | Loading [mol%] | Base | Solvent | Yield [%][b] |
|---|---|---|---|---|---|
| 1 | Cu(MeCN)$_4$BF$_4$ | 10 | DIPEA | CHCl$_3$ | 62 |
| 2 | CuOTf | 10 | DIPEA | CHCl$_3$ | 24 |
| 3 | CuI | 10 | DIPEA | CHCl$_3$ | 75 |
| 4 | CuBr | 10 | DIPEA | CHCl$_3$ | 81 |
| 5 | CuBr | 5 | DIPEA | CHCl$_3$ | 56 |
| 6 | CuBr | 20 | DIPEA | CHCl$_3$ | 79 |
| 7 | CuBr | 10 | – | CHCl3 | n.r. |
| 8 | CuBr | 10 | TEA | CHCl3 | 65 |
| 9 | CuBr | 10 | DMAP | CHCl3 | trace |
| 10 | CuBr | 10 | DBU | CHCl$_3$ | trace |
| 11 | CuBr | 10 | K$_2$CO$_3$ | CHCl$_3$ | 20 |
| 12 | CuBr | 10 | DIPEA | DCM | 75 |
| 13 | CuBr | 10 | DIPEA | toluene | 61 |
| 14 | CuBr | 10 | DIPEA | THF | 45 |
| 15 | CuBr | 10 | DIPEA | DMF | 48 |
| 16 | CuBr | 10 | DIPEA | H$_2$O | trace |
| 17[c] | CuBr | 10 | DIPEA | CHCl$_3$ | n.r. |
| 18[d] | CuBr | 10 | DIPEA | CHCl$_3$ | n.r. |

[a]Conditions: **1a** (1.0 equiv), **2a** (1.5 equiv), H$_2$O (1.0 equiv), base (2.0 equiv), HPO(OMe)$_2$ (0.1 equiv), solvent (0.1 M), room temperature for 6 h.
[b]Determined by $^1$H NMR of the crude mixture with an internal standard.
[c]Without HPO(OMe)$_2$.
[d]The reaction was set up without addition of water in absolutely anhydrous CHCl$_3$ with 4 Å MS.

inorganic base could only provide 20% yield of **3a** (Table 1, entry 11). The solvent effect was subsequently evaluated (Table 1, entries 12-16). The reaction could proceed smoothly in chloroform, dichloromethane (DCM), or toluene, whereas THF and DMF resulted in reducing yield with sluggish transformation (Table 1, entries 12–15). Only trace product could be observed using H$_2$O as solvent (Table 1, entry 16). Therefore, CHCl$_3$ was chosen as the most suitable solvent to afford desired *Z*-acrylamide. If HPO(OMe)$_2$ or water was removed from the system, almost no any reaction occurred under room temperature (Table 1, entries 17–18).

**Substrate scope for various alkynes**
With the optimized conditions in hand, we next examined the substrate scope for the MCR from alkynes aspect in Fig. 2. 20 kinds of OBoc-alkynes (**1**) were reacted with benzyl azide (**2a**) and H$_2$O to afford *Z*-acrylamides (**3**) in excellent selectivities (*Z:E* > 19:1). As shown in Fig. 2, both electron-donating and weakly electron-withdrawing OBoc-alkynes could participate in this transformation smoothly. Compared to 75% yield for non-substituted product **3a**, the yields of *p*-methoxyl and *p*-phenoxyl substrates were decreased to 69% and 60% for **3b** and **3c**. However, high yields were acquired as 71% (**3d**) and 76% (**3e**) for *p*-methylthio and *p*-methyl substituted substrates. Remarkably, more bulky substrate like *p-tert*-butyl group could be well tolerated and the yield was maintained at 60% (**3f**). For the OBoc-alkynes with weakly electron-withdrawing substituents like halogen, the acrylamides could be obtained in around 60–67% yields (**3g**-**3j**). Very similar yields (60–68%) were acquired for *m*-methyl, *m*-methoxy, *m*-chloro, *m*-bromo substituted OBoc-alkynes (**3k**-**3n**). But for *o*-methoxy and *o*-chloro substituted alkynes, the yields would drop to 56% and 54% may due to *ortho* effect (**3o**-**3p**). Noticeably, some complicated

*Z*-acrylamides (**3q**-**3t**) could also be gained from other aromatic or heterocyclic substrates, such as *α*-naphthyl, *β*-naphthyl, furyl and thienyl substituted substrates.

**Substrate scope for azides**
Then the substrate scope of different azides was screened in Fig. 3. Various *Z*-acrylamides (**3**) could be afforded in high stereospecificities (*Z:E* > 19:1) from alkyl or aryl azides. Both the electron-rich and electron-poor alkyl azides could provide *Z*-acrylamides with similar yields (70–74%), hinting that the electronic effect was not obvious for alkyl azides (**3a**, **3u**-**3v**). However, when phenyl azide was used instead of alkyl azides, the reaction became very sluggish and the yield would decline (**3w**). Encouragingly, good yields (**3x**-**3y**) were given for ethyl or butyl azides as substrates, which offered the potential opportunities for the further application in the synthesis of complicated molecules. In addition, we also investigated other types of aliphatic azides. Long-chain butyl-*di*-azide was used to successfully prepare butyl-azide-substituted *Z*-acrylamide product (**3z**), which provided the opportunity for the further application in polymerization using *di*-functional monomers. Cyclohexyl azide could efficiently participate in the transformation to obtain the corresponding *Z*-acrylamide product (**3aa**). Glycosyl azide could also be utilized to acquire carbohydrate derivate **3ab** in good yield, which offered a powerful way to access non-natural glycosyl alkenyl amide in glycomics study.

**Substrate scope for strongly electron-withdrawing alkynes**
When we explored the substrate range of alkynes, we were surprised to find that the strongly electron-withdrawing groups in alkynyl moieties can remarkably affect the chemoselectivities for this transformation, affording triazoles instead of predictable *Z*-acrylamides. Therefore, we

**Fig. 2 | Substrate scope for various alkynes[a].** [a]Conditions: **1** (1.0 equiv), **2a** (1.5 equiv), $H_2O$ (1.0 equiv), DIPEA (2.0 equiv), HPO(OMe)$_2$ (0.1 equiv), CHCl$_3$ (0.1 M), CuBr (10 mol%), room temperature for 6–12 h. [b]Yield of isolated product with $Z{:}E > 19{:}1$.

systematically evaluated the strongly electron-withdrawing effects in Fig. 4. When *p*-carbomethoxy OBoc-alkyne (**1ac**) was used as substrate, both *Z*-acrylamide (**3ac**) and OBoc-triazole (**4a**) were afforded in 20% and 51% yield respectively. When the alkynes substrates were substituted by much stronger electron-withdrawing groups such as cyano,

formyl and nitro were utilized in the reaction, only OBoc-triazoles as products were obtained in good yields and selectivities (**4b-4d**). The reason could be attributed to that such substrates disfavored the formation of alkylidene ketenimines because the strongly electron-withdrawing effect could destabilize the carbonium ions upon the

**Fig. 3 | Substrate scope for azides[a].** [a]Conditions: **1a** (1.0 equiv), **2** (1.5 equiv), $H_2O$ (1.0 equiv), DIPEA (2.0 equiv), HPO(OMe)$_2$ (0.1 equiv), CHCl$_3$ (0.1 M), CuBr (10 mol%), room temperature for 6–12 h. [b]Yield of isolated product with Z:E > 19:1.

removal of OBoc group, leading to the formation of OBoc-triazoles instead of Z-acrylamides. Therefore, the Cu-catalyzed azide-alkyne cycloaddition (CuAAC) reaction occurred to generate OBoc-triazoles as products instead of N$_2$ release-promoted rearrangement.

### The applications using *di*-OBoc-alkyne as substrate

It has already been demonstrated that *di*-azides could be well tolerated by such transformation to form corresponding Z-acrylamide (**3z**). We further investigated the applications to synthesize complicated molecules using *di*-OBoc-alkyne (**5a**) as substrate under the mild conditions in Fig. 5. It was unexpected to find that *p*-OBoc-triazolyl substituted Z-acrylamide (**3ad**) was afforded instead of Z-*di*-acrylamide (**3ae**), which encouraged us to explore the process in stepwise manner. In the first step, *p*-OBoc-alkynyl mono-substituted Z-acrylamide (**6a**) could be obtained via the MCR when controlling the feeding ratio of **5a**:**2a** to be 2:1. Then *p*-OBoc-triazolyl substituted Z-acrylamide (**3ad**) rather than **3ae** was acquired via CuAAC reaction. **6a** was considered as important intermediate for accessing triazolyl Z-acrylamide by inversing the electronic effect. The *p*-Z-acrylamide part in **6a** was a typical electron-withdrawing group to reverse the chemoselectivity for the transformation between OBoc-alkyne and azide, which was more favorable for the formation of triazole instead of Z-*di*-acrylamide. It is the first time to prepare the compound bearing both triazole and

Z-acrylamide starting from *di*-OBoc-alkyne in one step by in situ electronic effect inversion, which motivated us to extend above discoveries into MCP area.

### Multicomponent polymerization derived from MCR

Multicomponent polymerization (MCP) is a highly efficient polymerization strategy derived from corresponding MCR, rapidly affording functional polymers with diverse structures[7–21]. Based on the results using *di*-OBoc-alkyne, benzyl azides, and water, we further forwarded such transformation into MCP to prepare copolymers containing both acrylamide and triazole segments using *di*-OBoc-alkyne, *di*-azide, and water (Fig. 6). To verify the chemical structure of the polymers, model polymerization was initially performed using *di*-OBoc-alkyne (**5a**), *di*-benzyl azide (**7a**) and water to obtain **P1**. The chemical structure of **P1** was characterized and compared with the corresponding monomers (**5a** and **7a**) and small molecular model compound (**3ad**). As shown in Fig. 6, two sets of new peaks at δ 6.0 and 6.6 ppm representing the alkene protons, as well as the peak at δ 8.1 and 1.3 ppm representing the triazole and Boc protons emerged in such MCP. The spectrum of **P1** showed no peak associated with the resonances of the alkyne protons of **5a** at around 3.9 ppm, indicating that **5a** participated into such MCP. The protons of benzyl groups in **7a** (δ 4.5 ppm) notably shifted to 5.5 and 4.3 ppm in both spectra of **3ad**

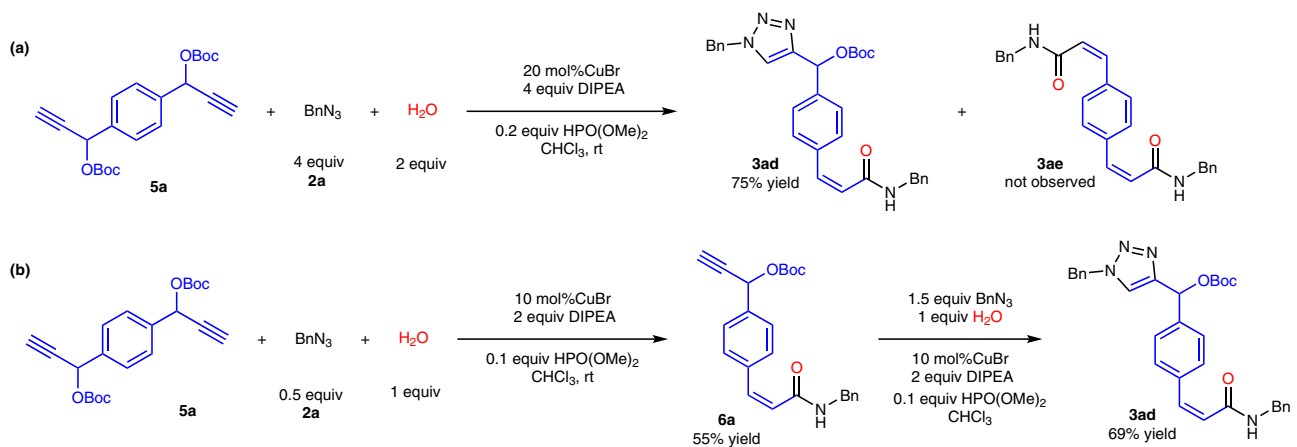

**Fig. 4 | Substrate scope for strongly electron-withdrawing alkynes[a].** [a]Conditions: **1** (1.0 equiv), **2a** (1.5 equiv), H₂O (1.0 equiv), DIPEA (2.0 equiv), HPO(OMe)₂ (0.1 equiv), CHCl₃ (0.1 M), CuBr (10 mol%), room temperature for 6–12 h. [b]Yield of isolated product.

**Fig. 5 | The applications using *di*-OBoc-alkyne as substrate. a 3ad** was afforded from MCR. **b 3ad** was afforded in stepwise manner.

and **P1**, demonstrating the change of chemical environment after the formation of acrylamide. Compared with the spectrum of model compound **3ad**, all other peaks can also be readily assigned as the similar resonance signals besides the typical peaks mentioned above. Moreover, It clearly showed that the integration of i′, b″ and g′ representing the triazole subunits were 1:1:2 (labeled as blue color), and the integration of l′, j′, k′, and m′ representing the acrylamide subunits were 1:1:1:2 (labeled as red color). The equal integration of typical protons in triazole and acrylamide revealed the exactly same proportion of both structures in the polymer, representing **P1** possibly owned the alternating subsets. Due to the mechanism for the step-growth polymerization, large amount of the **Int** bearing acrylamide was obtained at the first stage of the polymerization. Such **Int** also contained OBoc-alkyne and azide as two terminal groups. Based on the previous results, the CuAAC polymerization occurred between OBoc-alkyne and azide would lead to the formation of triazoles due to the existing of acrylamides as electron-withdrawing groups. During the polymerization, azide groups in **Int** could possibly react with **5a** as a

side-reaction, introducing defects in the alternating structures. However, such side-reaction could be ignored to a certain extent due to the dramatically different concentrations of azide groups between **7a** and **Int** in the system. As a result, **P1** with "approximate" alternating structure of triazole and acrylamide could be afforded via MCP. In order to further determine the alternating structure, the repeating unit information of the obtained polymer was analyzed using MALDI-TOF spectrum. **P2-OH** as a specific polymer with molecular weight ($M_n$) of 5000 g/mol was prepared using *di*-azides containing ether groups, followed by the transformation of -OBoc group to -OH group. A series of peaks with 490 Da separation were observed on the MALDI-TOF spectrum, consistent with the m/z of the **P2-OH**'s repeating unit (acrylamide-triazole) (Supplementary Fig. 13).

## Substrate scope for MCP

However, it was unsatisfied that the solubility of **P1** was a bit poor in high polar solvents like DMF and DMSO, which was possibly due to the high density of aromatic groups with rigidity. To further improve the

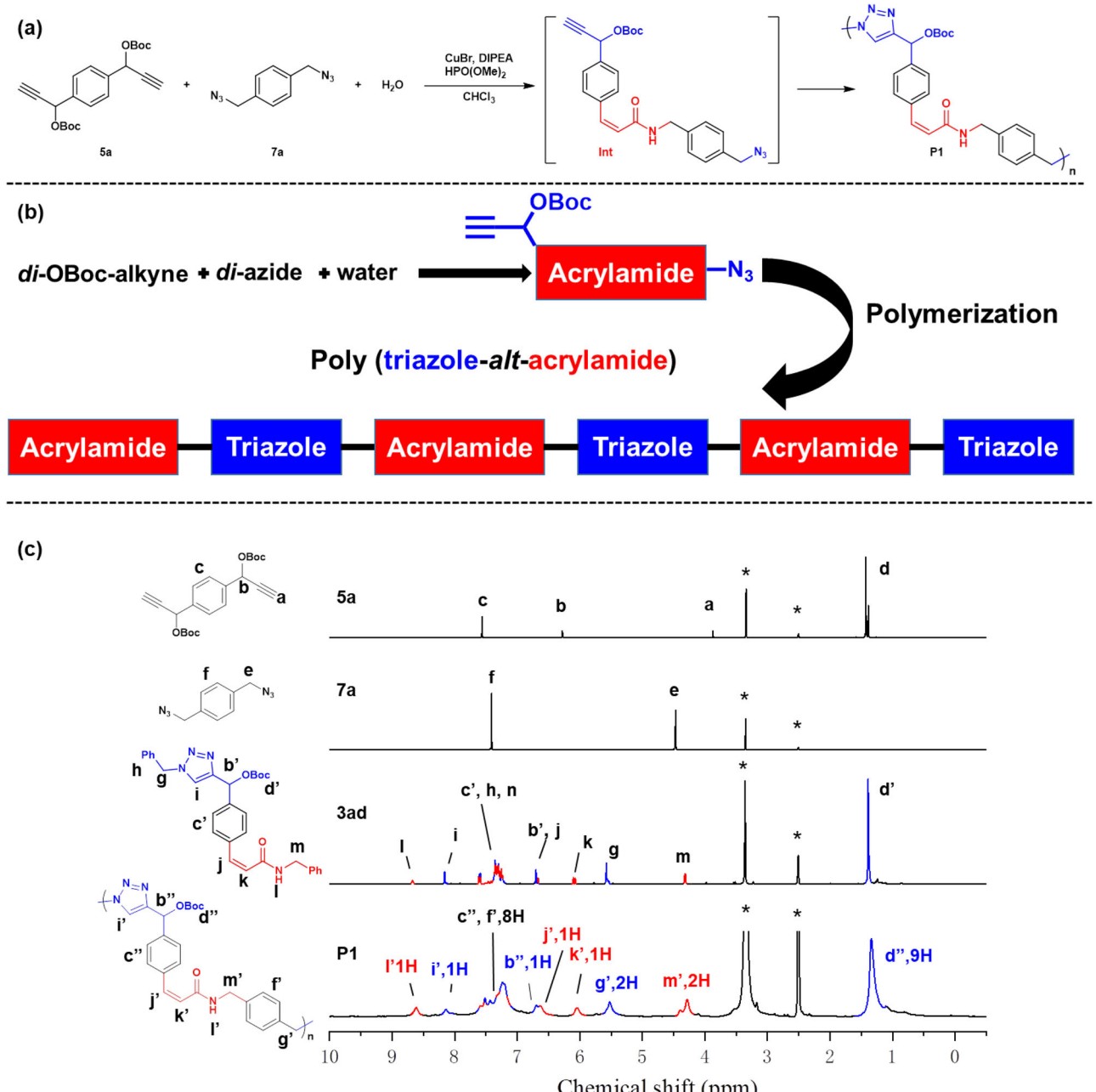

**Fig. 6 | MCP of *di*-OBoc-alkyne, *di*-azide, and water. a** Synthetic route of **P1**. **b** Schematic illustration of forming alternative polymer. **c** ¹H NMR spectra of **5a**, **7a**, **3ad**, and **P1** in $d_6$-DMSO. The solvent peaks including $d_6$-DMSO and H₂O were marked as asterisks.

solubility of the polymers and expand the MCP scope, various *di*-azides with flexible aliphatic chains or ether groups were introduced into such MCP (Fig. 7). It was delight to find that all the MCP proceeded smoothly to afford 5 kinds of high molecular weight ($M_n$) polymers ($M_n$ up to 44300 g/mol) with high yields. The solubility of the polymers has been significantly improved by enhancing the flexibility of the azides. Polymers bearing multi-ether groups (**P2** and **P4**) could be easily dissolved in most of organic solvents such as chloroform and THF. The ¹H NMR spectra of all the polymers also convinced the equal portion of triazole and acrylamide in the polymers.

### Proposed mechanism for the synthesis of *Z*-acrylamide

On the basis of various mechanistic experiments (Supplementary Figs. 1–9) and literatures[41–46], we proposed the reaction mechanism in Fig. 8. OBoc-alkyne **1a** initially coordinates with copper(I) to form

copper acetylide **A** in the presence of base. Subsequently, the -OBoc group is removed by an $S_N1$ mechanism to form cationic intermediate **B**, which further generates copper-allenylidene complex **B'** as a resonance structure[41–45]. Then the azide-Cu-allenylidene [3 + 2] cycloaddition reaction occurs to form intermediate **C** with allenyl copper[46]. Copper would be easily eliminated may due to the assist with phosphite from the unstable intermediate **C** to give carbene intermediate **D**, which is further trapped by GC-MS (gas chromatography-mass spectrometry) with HRMS ($m/z = 249.1259$) (Supplementary Figs. 5–8). A carbene-assisted rearrangement occurs with extrusion of N₂ to afford alkylidene ketenimine **E**. Alkylidene ketenimine **E** is undoubtedly confirmed by GC-MS with HRMS ($m/z = 219.1042$). Then, Cu(I) would coordinate with **E** to form intermediate **F**. According to mechanistic experiments using isotope labeling (Supplementary Figs. 2, 3, and 4), water could *cis*-add to the middle double bond of **F** to give the imino

**Fig. 7 | Substrate scope for MCP[a].** [a]Conditions: **5a** (1.0 equiv), **7** (1.0 equiv), H$_2$O (2.0 equiv), DIPEA (4.0 equiv), HPO(OMe)$_2$ (1.0 equiv), CHCl$_3$ (0.1 M), CuBr (20 mol%), room temperature for 12–24 h. [b]$M_n$, and Đ were determined by GPC in DMF with PMMA standards.

acid **3a'** due to the lone pair-$\pi$ interaction between nitrogen and aromatic rings supported by the density functional theory (DFT) calculations (Supplementary Figure 9). *Z*-acrylamide compound **3a** is obtained as product from an isomerization process of **3a'**.

In summary, we have developed an unexpected three-component reaction of alkyne, azide and water by the cascade cycloaddition, N$_2$ extrusion and carbene-assisted rearrangement involving in situ formation of alkylidene ketenimine intermediate under mild conditions. Only *Z*-$\alpha,\beta$-unsaturated amides instead of *E*-$\alpha,\beta$-unsaturated amides or triazoles were afforded from alkylidene ketenimines with excellent selectivities and stereospecificities (*Z:E* > 19:1). This method owning broad substrate scope (36 examples) could be potentially used to prepare bioactive *Z*-selective $\alpha,\beta$-unsaturated amides, which were very

difficult to access before. In addition, a series of unique poly (triazole-*alt*-*Z*-acrylamides) with high $M_n$s and yields were efficiently afforded by MCP through a very simple operation basing on this MCR. Therefore, we believe that this work will arouse the attentions and motivate more synthetic methodologies involving alkylidene ketenimines in future.

## Methods

### Representative procedures for the synthesis of *Z*-acrylamides

OBoc-alkyne **1a** (46.4 mg, 0.2 mmol, 1.0 equiv) was added to a vial containing CHCl$_3$ (2 mL). Then DIPEA (66 μL, 2.0 equiv), HPO(OMe)$_2$ (2 μL,10 mol %), H$_2$O (3.6 μL, 1.0 equiv), CuBr (3.8 mg, 10 mol%) and benzyl azide **2a** (37.5 μL, 1.5 equiv) were added to the mixture. The vial was closed and the mixture was stirred for 6 h at room temperature.

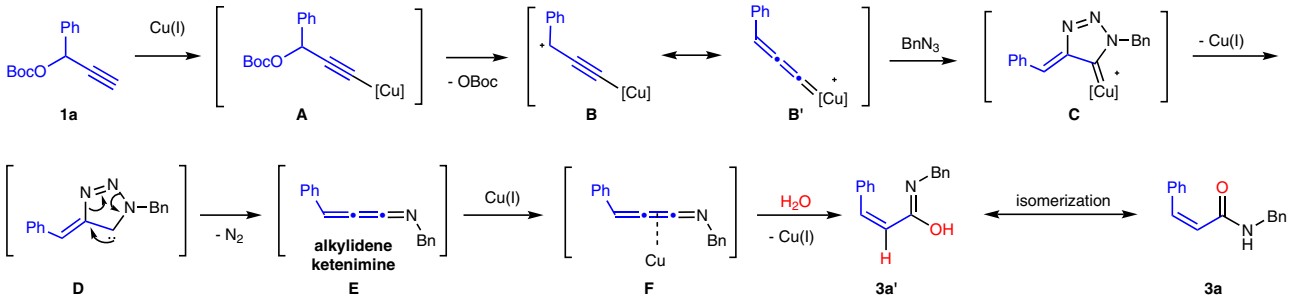

**Fig. 8 | Proposed mechanism for the synthesis of *Z*-acrylamide.** The reaction proceeds through a carbene-assisted rearrangement to form alkylidene ketenimine intermediate.

After the solvent was removed under reduced pressure, the residue was purified with flash column chromatography (33% EtOAc in petroleum ether) to give product **3a** (36 mg, 75% yield) as a yellow solid.

## Data availability

All data generated and analyzed in this paper are available in the Supplementary Information. Experimental procedures, characterization of new compounds, and DFT calculations (Supplementary Data 1 and Supplementary Data 2 for Cartesian coordinates) are available in the Supplementary information.

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

## Acknowledgements

This work was supported by grants from the National Natural Science Foundation of China (21978039) (to W. S.), Natural Science Foundation of Jiangsu Province (BK20211100) (to N. Z.), the Fundamental Research Funds for the Central Universities (DUT21YG133) (to N. Z.) and (DUT22YG224) (to W. S.).

## Author contributions

W.S. and N.Z. designed the study. X.D., M.L., G.L., X.S., and Q.W. performed the experiments. X.D and M.L. analyzed the data. X.D., W.S. and N.Z. wrote the manuscript. All authors discussed the results and commented on the manuscript.

## Competing interests

The authors declare no competing interests.
