## [Peer Review File · Nature Communications]

Copper-Catalyzed Formation of Alkylidene Ketenimines: Z-Selective Synthesis of Acrylamides and Poly-AcrylamidesREVIEWER COMMENTS

Reviewer #1 (Remarks to the Author):

In this manuscript, Song and co-workers developed a copper catalyzed tandem N₂ extrusion and carbene-assisted rearrangement reaction. This strategy provides a rapid access to various Z-acrylamides and poly-acrylamides. It is an interesting work, however, some limitations remain in this method. Therefore, it is acceptable for after major revision.

1. In the reaction, water acts as a nucleophilic reagent and proton source. Whether other nucleophiles, such as alcohols or primary amines, could perform the transformation.
2. The alkynes of aryl or hetero-aryl group worked well, whether alkyl substituted ones could perform the reaction.
3. HPO(OMe)₂ seems to be critical to the reaction, but discussion about the choose of HPO(OMe)₂ is omitted, and its function is not discussed in the proposed mechanism.
4. Cis-addition of alkylidene ketenimine with water was proposed to explain the Z-selectivity. More details and control experiments is suggested, such as using D₂O instead of H₂O, excessive amount of water to Z/E selectivity.
5. The yields ranged between 60% and 70% for the products, please indicate the main byproducts or the reasons for the moderate yield.
6. The reaction condition for cycloaddition to generate trizaoles is as same as the acrylamides. Is the addition of water and HPO(OMe)₂ necessary? The total yield for sequential synthesis of 3ad (Fig 5) is moderate, but up to 99% yield was obtained in the MCP transformation of synthesis of P. Please rationalize the improved yield.

Other mistakes:

1. The loading should be "mol%" instead of "%".
2. The Z/E selectivity should be added in the Table or footnote, and how dose the ratio is determined. In addition, >99:1 Z/E radio is not rigorous, and usually 95:5 is the limitation in H NMR determination.
3. Please check the general structures of the products in Fig 3, Fig 4,
4. In Fig 5 and others, eq should be equiv.
5. The product number in this paper is confused and needs to be unified, such as '3aa, 3ab....' and '3ba, 3ca....'.
6. The use of organic or inorganic copper salt is not appreciate.

Some corrections in supporting information:

1. Please correct the ¹³C NMR spectrum analysis of fluorine-containing compound 3g, and add the ¹⁹F NMR spectrum of 3g.
2. Please indicate isotopic peaks of chlorine-containing compounds and bromine-containing compounds in HRMS.

Reviewer #2 (Remarks to the Author):

This work reported the copper-catalyzed multicomponent reaction of alkyne, azide, and water to synthesis Z-acrylamides selectively. The mechanism of this reaction and the difference of reactivity of the bifunctional alkyne monomer before and after the formation of one acrylamide unit were revealed. The corresponding multicomponent polymerizations were also developed. The work is generally interesting and well-organized. However, to meet the high standard of Nature Communication, the following questions should be fully addressed.

1. The synthesis of Z-acrylamide isomer may be difficult, but the importance for such regioselective structure should also be discussed. What is the difference of E- and Z-isomers could we expect in terms of their property?
2. The manuscript did not provide enough explanation about how the E/Z stereoselectivity generated. In the proposed mechanism, the addition of water on the alkylidene ketenimine may not be strictly stereoselective.
3. The reactions in this literature generally possess yields of 50-76%, which are not very high. What did the rest reactants transform to?

4. One of the main claims about this manuscript is that because “3ad was formed instead of 3ae”, polymers with alternating triazole and acrylamide structures were formed. However, this strict alternating structure has not been convincingly proved. First of all, the main evidence about this alternating structure is the “equal integration of typical protons in triazole and acrylamide”, however, the authors did not provide the integral data on the ¹H NMR spectra of the polymers. Secondly, in Fig 6, the proton peaks for g' and m' should be strictly 1:1 ratio, however, there are clear shoulder peaks emerged for m', indicating more complicated polymer structures. Most importantly, even if Int 7 was formed first and the remaining alkyne group could only transform to triazole, the remaining azide group could still react with 5a to form acrylamide structures, leading to random structures. The authors should carefully re-consider the polymer structures.

5. Other discuss should also be provided:

(a) What is the role of HPO(OMe)₂ in this reaction?

(b) Why Boc group is necessary in the substrates? How about replacing it with other protecting groups?

(c) Please provide the original ¹H and ¹³C NMR spectra of compound 3ad in DMSO-d₆ in supporting information.

6. There are some up-to-date literatures documented the multicomponent polymerizations involving alkyne and sulfonyl azide monomers. The related literature should be thoroughly checked and cited.

7. There are a few typos existed in the manuscript. For example, line 147, “formaton”; I do not think the structure of “Int 7” could be claimed as a “dimer”; no “Scheme 3” found in the manuscript.

To Reviewer 1:

Comments:

In this manuscript, Song and co-workers developed a copper catalyzed tandem N2 extrusion and carbene-assisted rearrangement reaction. This strategy provides a rapid access to various Z-acrylamides and poly-acrylamides. It is an interesting work, however, some limitations remain in this method. Therefore, it is acceptable for after major revision.

1. In the reaction, water acts as a nucleophilic reagent and proton source. Whether other nucleophiles, such as alcohols or primary amines, could perform the transformation.

Our response:

We have examined other nucleophiles such as alcohols, amines and mercaptans and summarized as “8. Examination of other nucleophiles” in Supporting Information. However, only trace product was observed in crude ¹H-NMR spectrum using benzyl alcohol as nucleophile (Scheme S4a). We have tried to improve the yields under various reaction conditions but failed. When primary or secondary amines were used as nucleophiles, corresponding amino-triazoles **S1** and **S2** were obtained instead of acrylamidines due to the strong nucleophilicity of amine groups (Scheme S4b). If benzyl mercaptan was used as nucleophile, only unknown insoluble solid was acquired which was undissolved in almost all organic solvents and water (Scheme S4c). Therefore, stronger nucleophiles may not be well tolerated for this transformation.

Scheme S4. The examination of other nucleophiles

2. The alkynes of aryl or hetero-aryl group worked well, whether alkyl substituted ones could perform the reaction.

Our response:

We have tried alkyl substituted alkynes such as *tert*-butyl hept-1-yn-3-yl carbonate for this transformation, however, the reaction was very messy with acquiring unknown mixture, and no any desired product was isolated. Therefore, the aryl or hetero-aryl group is necessary for this transformation. It is possible because the aromatic structures could stabilize the cationic intermediate **B** after the removal of -OBoc group (see Fig 8 in the manuscript).

3. HPO(OMe)₂ seems to be critical to the reaction, but discussion about the choose of HPO(OMe)₂ is omitted, and its function is not discussed in the proposed mechanism.

Our response:

We have finished several control experiments to figure out the role of HPO(OMe)₂ and summarized in Scheme S2 in the Supporting Information. First, if no dimethyl

phosphite was added, trace desired acrylamide was observed from crude $^1\text{H-NMR}$ spectrum (Scheme S2a). Second, if azide was removed from the system, only allenyl phosphonate was isolated instead of acrylamide. This transformation has already been reported (Chem. Commun. **2016**, 52, 6451). Treating allenyl phosphonate by azide in standard reaction conditions, no any reactions occurred (Scheme S2b). Third, if alkyne was removed from system, no any reactions occurred (Scheme S2c). Finally, we prepared OBoc-triazole as substrate by other method. Treating OBoc-triazole by azide and water in standard reaction conditions, no any reactions occurred (Scheme S2d).

(a) no $\text{HPO}(\text{OMe})_2$

(b) no azide

(c) no alkyne

(d) OBoc-triazole as substrate

Scheme S2. The control experiments

Subsequently, after screening various phosphites, we found that only dimethyl phosphite ($\text{HPO}(\text{OMe})_2$) and diethyl phosphite ($\text{HPO}(\text{OEt})_2$) could promote this transformation (75% yield for dimethyl phosphite and 60% yield for diethyl phosphite). However, the yield for diisopropyl phosphite $\text{HPO}(\text{iPr})_2$ would be dramatically decreased to 30%. Therefore, we have considered that dimethyl phosphite may be as important assisted nucleophile or ligand with copper, which

could promote the copper elimination from unstable intermediate **C** to generate carbene intermediate **D** as follows (Scheme S3):

Scheme S3. The details for phosphite-assist carbene formation

Next, ^{31}P -NMR spectra were used to monitor this transformation in Figure S5. The chemical shift of $\text{HP}(\text{OMe})_2$ was δ 10.44 ppm in CDCl_3 . After we set up the reaction in small scales in NMR tube using CDCl_3 as solvent, a new peak was observed in δ -17.72 ppm within 10 min. We considered that the intermediate $[\text{Cu}]\text{-P}(\text{OH})(\text{OMe})_2$ could be observed in -17.72 ppm. After 1 hour, we could still observed the major peaks in δ 10.44, -17.73 ppm, which could maintain unchangeably during the whole reaction process. Above ^{31}P -NMR spectra could support the proposed mechanism in Scheme S3.

Figure S5. ^{31}P -NMR spectra for monitoring the transformation

(a) ^{31}P -NMR spectrum of $\text{HP}(\text{OMe})_2$

(b) ^{31}P -NMR spectrum of reaction mixture within 10 min

(c) ^{31}P -NMR spectrum of reaction mixture after 1 hour

We also mentioned the role of $\text{HPO}(\text{OMe})_2$ in the manuscript as follows:

“Copper would be easily eliminated may due to the assist with phosphite from the unstable intermediate **C** to give carbene intermediate **D**, which is further trapped by GC-MS (gas chromatography-mass spectrometry) with HRMS ($m/z = 249.1259$) (Scheme S2, Scheme S3, Figure S4, Figure S5).”

4. Cis-addition of alkylidene ketenimine with water was proposed to explain the *Z*-selectivity. More details and control experiments is suggested, such as using D₂O instead of H₂O, excessive amount of water to *Z/E* selectivity.

Our response:

The effect of water amount was systematically investigated in Figure S1 in the Supporting Information. When the amount of H₂O was controlled less than 5 equiv, the yield did not change significantly. In addition, the excessive amount of water could not obviously influence *Z/E* selectivity.

Figure S1. The effect of water equivalent on the reaction

Equiv	Yield(%)	Z:E
1	75	>19:1
2	75	>19:1
5	72	>19:1

Z-acrylamides were afforded by adding water to alkylidene ketenimine intermediates. When D₂O or H₂¹⁸O were used instead of H₂O, the results were showed in Scheme S1.

Scheme S1. The investigation for mechanism

To further investigation the details of the mechanism, two mechanistic experiments were designed in Scheme S1. First, D₂O was used as a reactant. According to the ¹H NMR spectra, the hydrogen on the olefin double bond was partially replaced by deuterium (see Figure S2). Next, heavy-oxygen water (H₂¹⁸O) was used instead of H₂O, and the ¹⁸O substituted product was successfully characterized by high resolution mass spectrometry (HRMS) as m/z (M+Na)⁺ 262.1088 (see Figure S3). It demonstrated that the water could *cis*-add to the middle double bond of **E** (see Fig. 8 in the manuscript) to give the imino acid **3a'**, which quickly isomerized to Z-acrylamide compound **3a**.

Figure S2. ¹H NMR spectra of deuterium Z-acrylamide

Figure S3. HRMS of ^{18}O substituted Z-acrylamide

SWZ

2021102100 21 (0.431) AM2 (Ar,20000.0,556.28,0.00,LS 10)

We have modified the mechanism as follows to show the formation of Z-selective products clearly and in details (Fig. 8). Cu(I) would coordinate with **E** to form intermediate **F**. According to mechanistic experiments (see the Supporting Information), water could *cis*-add to the middle double bond of **F** to give the imino acid **3a'** due to the lone pair- π interaction between nitrogen and aromatic rings supported by the density functional theory (DFT) calculations (Figure S6). Z-acrylamide compound **3a** is obtained as product from an isomerization process of **3a'**.

The mechanism and selectivity were further discussed in the Supporting Information. Intermediate **F** could be trapped by MALDI-TOF (see Figure S7). The Z-selectivity was determined in the formation of imino acid **3a'**. We have considered that the lone pair from nitrogen may contribute to the formation of Z-selectivity by the lone pair- π interaction (see Figure S6). We have calculated the stable configuration of **3a'** and **3a''** by density functional theory (DFT) calculations performing at the M06/6-31G(d)//B3LYP/6-31G(d)-LANL2DZ level. The distance from nitrogen atom to aryl group is about 3.2 Å, which is suitable for the lone pair- π interaction. It also

may be the reason why the reaction would become very messy without any selectivity using alkyl substituted groups instead of aryl groups in alkyne moieties.

Figure S6. The stable configurations of imino acids

5. The yields ranged between 60% and 70% for the products, please indicate the main byproducts or the reasons for the moderate yield.

Our response:

The starting materials were still left without fully converting in 12 h, which may result in moderate yields for some cases. In addition, the allenyl phosphonates (see Scheme S2b) could be isolated under Cu(I)-catalyzed base conditions as byproducts in some cases, which may be another reason for the moderate yields.

6. The reaction condition for cycloaddition to generate triazoles is as same as the acrylamides. Is the addition of water and HPO(OMe)₂ necessary? The total yield for sequential synthesis of 3ad (Fig 5) is moderate, but up to 99% yield was obtained in the MCP transformation of synthesis of P. Please rationalize the improved yield.

Our response:

According to the results showed in Fig. 4, triazoles could only be acquired using strongly electron-withdrawing substituent groups such as cyano, formyl and nitro alkynes. We found that nitro substituted triazole **4d** could also be afforded in 84% yield without water and HPO(OMe)₂. Therefore, it is not necessary to use water and HPO(OMe)₂ for the preparation of electron-withdrawing groups substituted triazoles. Due to the residual copper left in the polymer, the over high yields were obtained. After washing the polymers with EDTA for many times to make sure no copper left,

the adjusted yields were acquired in the manuscript as “P1: 83%, P2: 87%, P3: 82%, P4: 85%, P5: 88%”.

Other mistakes:

1. The loading should be “mol%” instead of “%”.

Our response:

We have revised the loading as “mol%” instead of “%” in the manuscript.

2. The Z/E selectivity should be added in the Table or footnote, and how dose the ratio is determined. In addition, >99:1 Z/E radio is not rigorous, and usually 95:5 is the limitation in H NMR determination.

Our response:

The Z/E selectivity was added in the Table. The Z/E selectivity was determined by ¹H NMR of the crude mixture. We have revised the Z/E radio as “> 19:1” instead of “> 99:1” in the manuscript.

3. Please check the general structures of the products in Fig 3, Fig 4.

Our response:

We have revised the general structures of the products in Fig 3 and Fig 4.

4. In Fig 5 and others, eq should be equiv.

Our response:

We have revised the “eq” to “equiv” in Fig. 5 and others in the manuscript.

5. The product number in this paper is confused and needs to be unified, such as ‘3aa, 3ab....’ and ‘3ba, 3ca....’.

Our response:

In this manuscript, many different reactions and products were involved. Hence, we have tried our best to avoid confusing for giving the unified numbers for each compound. The revised illustrations for compound number are as follows:

We named acrylamides as “3a, 3b, ... 3aa, 3ab, 3ac, 3ad, and 3ae”.

We named triazoles for strongly electron-withdrawing substituted products as “4a, 4b, 4c, and 4d”.

We named *di*-OBoc-alkyne substrate as “5a” and the corresponding product has been re-named as “6a”.

We re-named *di*-benzyl azide as “7a” and revised azide as “7” in Fig. 6 and Fig. 7.

We have revised the “Int 7” to “Int” to avoid misunderstanding.

We named polymers as “P1, P2, ... P5”.

6. The use of organic or inorganic copper salt is not appreciate.

Our response:

We have deleted “organic or inorganic copper” and modified the contents as “A series of commercially available coppers(I) were investigated. ... To our delight, CuI and CuBr were demonstrated more efficient than Cu(MeCN)₄BF₄ and CuOTf for this reaction (Table 1, entries 3-4).” in the manuscript.

Some corrections in supporting information:

1. Please correct the ¹³C NMR spectrum analysis of fluorine-containing compound 3g, and add the ¹⁹F NMR spectrum of 3g.

Our response:

We have corrected ¹³C NMR spectrum and ¹⁹F NMR spectrum of 3g in Supporting Information as follows:

¹³C NMR (100 MHz, CDCl₃) δ 166.6, 164.00, 161.5, 137.6, 135.9, 131.2 (d, *J* = 8.2 Hz), 128.7, 128.0, 127.6, 124.1, 115.3 (d, *J* = 21.6 Hz), 43.6. ¹⁹F NMR (565 MHz, CDCl₃) δ 112.2.

2. Please indicate isotopic peaks of chlorine-containing compounds and bromine-containing compounds in HRMS.

Our response:

We have added isotopic peaks of chlorine-containing compounds and bromine-containing compounds in Supporting Information.

To Reviewer 2:

This work reported the copper-catalyzed multicomponent reaction of alkyne, azide, and water to synthesis Z-acrylamides selectively. The mechanism of this reaction and the difference of reactivity of the bifunctional alkyne monomer before and after the

formation of one acrylamide unit were revealed. The corresponding multicomponent polymerizations were also developed. The work is generally interesting and well-organized. However, to meet the high standard of Nature Communication, the following questions should be fully addressed.

1. The synthesis of Z-acrylamide isomer may be difficult, but the importance for such regioselective structure should also be discussed. What is the difference of E- and Z-isomers could we expect in terms of their property?

Our response:

We have mentioned the unique properties for Z-isomer as “Compared with E-isomer, the Z-selective acrylamides could be widely used as unique building blocks for synthetic transformations and drug delivery, or as the important structures for bioactive molecules such as motualevic acid B and basiliskamide A.³³⁻³⁷” in the manuscript. We also added more relative references 33-36 as follows:

33. Zhang, S., Neumann, H. & Beller, M. Synthesis of α,β -Unsaturated Carbonyl Compounds by Carbonylation Reactions. *Chem. Soc. Rev.* **49**, 3187-3210 (2020).

34. Wang, W., Camenisch, G., Sane, D. C., Zhang, H., Hugger, E., Wheeler, G. L., Borchardt, R. T. & Wang, B. A Coumarin-Based Prodrug Strategy to Improve the Oral Absorption of RGD Peptidomimetics. *J. Control. Release.* **65**, 245-251 (2000).

35. Keffer, J. L., Plaza, A. & Bewley, C. A. Motualevic Acids A–F, Antimicrobial Acids from the Sponge *Siliquariaspongia* sp. *Org. Lett.* **11**, 1087-1090 (2009).

36. Barsby, T., Kelly, M. T. & Andersen, R. J. Tupuseleiamides and Basiliskamides, New Acyldipeptides and Antifungal Polyketides Produced in Culture by a *Bacilluslaterosporus* Isolate Obtained from a Tropical Marine Habitat. *J. Nat. Prod.* **65**, 1447-1451 (2002).

2. The manuscript did not provide enough explanation about how the E/Z stereoselectivity generated. In the proposed mechanism, the addition of water on the alkylidene ketenimine may not be strictly stereoselective.

Our response:

We have modified the mechanism as follows to show the formation of Z-selective products clearly and in details (Fig. 8). Cu(I) would coordinate with **E** to form intermediate **F**. According to mechanistic experiments (see the Supporting Information), water could *cis*-add to the middle double bond of **F** to give the imino acid **3a'** due to the lone pair- π interaction between nitrogen and aromatic rings supported by the density functional theory (DFT) calculations (Figure S6). Z-acrylamide compound **3a** is obtained as product from an isomerization process of **3a'**.

The mechanism and selectivity were further discussed in the Supporting Information. Intermediate **F** could be trapped by MALDI-TOF (see Figure S7). The *Z*-selectivity was determined in the formation of imino acid **3a'**. We have considered that the lone pair from nitrogen may contribute to the formation of *Z*-selectivity by the lone pair- π interaction (see Figure S6). We have calculated the stable configuration of **3a'** and **3a''** by density functional theory (DFT) calculations were performed at the M06/6-31G(d)//B3LYP/6-31G(d)-LANL2DZ level. The distance from nitrogen atom to aryl group is about 3.2 Å, which is suitable for the lone pair- π interaction. It also may be the reason why the reaction would become very messy without any selectivity using alkyl substituted groups instead of aryl groups in alkyne moieties.

Figure S6. The stable configurations of imino acids

3. The reactions in this literature generally possess yields of 50-76%, which are not very high. What did the rest reactants transform to?

Our response:

The starting materials were still left without fully converting in 12 h, which may result in moderate yields for some cases. In addition, the allenyl phosphonates (see Scheme S2b) could be isolated under Cu(I)-catalyzed base conditions as byproducts in some cases, which may be another reason for the moderate yields.

4. One of the main claims about this manuscript is that because “3ad was formed instead of 3ae”, polymers with alternating triazole and acrylamide structures were formed. However, this strict alternating structure has not been convincingly proved. First of all, the main evidence about this alternating structure is the “equal integration of typical protons in triazole and acrylamide”, however, the authors did not provide the integral data on the ^1H NMR spectra of the polymers. Secondly, in Fig 6, the proton peaks for g' and m' should be strictly 1:1 ratio, however, there are clear shoulder peaks emerged for m', indicating more complicated polymer structures. Most importantly, even if Int 7 was formed first and the remaining alkyne group could only transform to triazole, the remaining azide group could still react with 5a to form acrylamide structures, leading to random structures. The authors should carefully re-consider the polymer structures.

Our response:

First, we have provided the integral information on the ^1H NMR spectra of the polymers in the revised **Figure 6**. It clearly showed that the integration of i' and g' representing the triazole subunits were 1:2 (labeled as blue color), and the integration of l', j', k', and m' representing the acrylamide subunits were 1:1:1:2 (labeled as red color).

Fig 6. ^1H NMR spectra of **5a** (A), **7a** (B), **3ad** (C), and **P1** (D) in d_6 -DMSO

Second, the integration of proton peaks for g' and m' were strictly 1:1 even m' was a shoulder peak. It was possible because of the *cis-trans* isomerism of amide groups. In general, the *cis-trans* isomerism of amide groups could not be observed at room temperature for small molecules due to the similar energy barrier (~ 10.5 kJ/mol). However, such *cis-trans* isomerism of amide groups could be possibly observed when the molecules were hard to rotate, for example, in the polymers or with bulky substituted groups, which may result in a shoulder peak of m' in the polymers.

Third, in order to determine the alternating structure, MALDI-TOF analysis of **P2-OH** was performed as below. Since large molecular weight polymers were hard to characterized using MALDI-TOF, **P2-OH** with low Mw (about 5000 g/mol) was prepared followed by the transformation of -OBoc group to -OH group (No signal was found from MALDI-TOF results using **P2** with OBoc groups). A series of peaks with 490 Da separations were observed in the MALDI-TOF spectrum, consistent with the m/z of the **P2-OH**'s repeating unit (acrylamide-triazole). No separations belong to polytriazoles or polyacrylamides were found from the MALDI-TOF spectrum, demonstrating the alternating structure of the polymers (see Figure S7).

Figure S7. MALDI-TOF spectrum and analysis of **P2-OH**.

Measured (g/mol)	Exact Mass (expected, g/mol)	End-group
475	472	
505	502	
533	535	
967	962	
995	995	
1024	1025	
1457	1452	
1485	1482	
1513	1515	

In addition, at the early stage of this polymerization, considering the large concentration differences of the azide group in **7a** and **Int**, **Int** was quite hard to react with **5a** since **5a** was more favorable to react with **7a** instead of **Int**. During the

consecutive consumption of the monomers, it is true that the azide group in **Int** could possible react with **5a** to form acrylamide structures since the concentration of **7a** was much lower than **Int** in the system. However, such side-reaction could be ignored since the concentration of **5a** was also quite low (the same with **7a**). Actually, it may not be ideal alternative polymers due to the existence of the above-mentioned side-reaction. Therefore, strictly speaking, we changed the description of polymers to be “approximate alternating polymers” in the manuscript.

5. Other discuss should also be provided:

(a) What is the role of HPO(OMe)₂ in this reaction?

Our response:

We have finished some control experiments to figure out the role of HPO(OMe)₂ and summarized in Supporting Information. First, if no dimethyl phosphite was added, trace desired acrylamide was observed from crude ¹H-NMR spectrum (Scheme S2a). Second, if azide was removed from the system, only allenyl phosphonate was isolated instead of acrylamide. This transformation has already been reported (Chem. Commun. **2016**, 52, 6451). Treating allenyl phosphonate by azide in standard reaction conditions, no any reactions occurred (Scheme S2b). Third, if alkyne was removed from system, no any reactions occurred (Scheme S2c). Finally, we prepared OBoc-triazole as substrate by other method. Treating OBoc-triazole by azide and water in standard reaction conditions, no any reactions occurred (Scheme S2d).

(a) no HPO(OMe)₂

(b) no azide

(c) no alkyne

(d) OBoc-triazole as substrate

Scheme S2. The control experiments

Subsequently, after screening various phosphites, we found that only dimethyl phosphite (HPO(OMe)₂) and diethyl phosphite (HPO(OEt)₂) could promote this transformation (75% yield for dimethyl phosphite and 60% yield for diethyl phosphite). However, the yield for diisopropyl phosphite HPO(ⁱPr)₂ would be dramatically decreased to 30%. Therefore, we have considered that dimethyl phosphite may be as important assisted nucleophile or ligand with copper, which could promote the copper elimination from unstable intermediate **C** to generate carbene intermediate **D** as follows (Scheme S3):

Scheme S3. The details for phosphite-assist carbene formation

Next, ^{31}P -NMR spectra were used to monitor this transformation in Figure S5. The chemical shift of $\text{HP}(\text{OMe})_2$ was δ 10.44 ppm in CDCl_3 . After we set up the reaction in small scales in NMR tube using CDCl_3 as solvent, a new peak was observed in δ -17.72 ppm within 10 min. We considered that the intermediate $[\text{Cu}]\text{-P}(\text{OH})(\text{OMe})_2$ could be observed in -17.72 ppm. After 1 hour, we could still observed the major peaks in δ 10.44, -17.73 ppm, which could maintain unchangeably during the whole reaction process. Above ^{31}P -NMR spectra could support the proposed mechanism in Scheme S3.

Figure S5. ^{31}P -NMR spectra for monitoring the transformation

(a) ^{31}P -NMR spectrum of $\text{HP}(\text{OMe})_2$

(b) ^{31}P -NMR spectrum of reaction mixture within 10 min

(c) ^{31}P -NMR spectrum of reaction mixture after 1 hour

We also mentioned the role of $\text{HPO}(\text{OMe})_2$ in the manuscript as follows:

“Copper would be easily eliminated may due to the assist with phosphite from the unstable intermediate **C** to give carbene intermediate **D**, which is further trapped by GC-MS (gas chromatography-mass spectrometry) with HRMS ($m/z = 249.1259$) (Scheme S2, Scheme S3, Figure S4, Figure S5).”

(b) Why Boc group is necessary in the substrates? How about replacing it with other protecting groups?

Our response:

We have screened different leaving groups and Boc group was demonstrated to be necessary for this transformations. If Ac and Piv groups were used instead of Boc group, only triazoles **S3** and **S4** were afforded with 85% and 83% yields respectively instead of acrylamides under reaction conditions (Scheme S5).

Scheme S5. The examination of leaving groups

We have added above results in the Supporting Information as “9. Examination of leaving groups”.

(c) Please provide the original ^1H and ^{13}C NMR spectra of compound 3ad in DMSO- d_6 in supporting information.

Our response:

We have added ^1H and ^{13}C NMR spectra of compound 3ad in d_6 -DMSO in Supporting Information.

^1H NMR (400 MHz, d_6 -DMSO) δ 8.67 (s, 1H), 8.16 (s, 1H), 7.60 (d, $J = 8.1$ Hz, 2H), 7.40 – 7.24 (m, 12H), 6.68 (d, $J = 14.4$ Hz, 2H), 6.09 (d, $J = 12.8$ Hz, 1H), 5.58 (s, 2H), 4.32 (d, $J = 5.9$ Hz, 2H), 1.39 (s, 9H). ^{13}C NMR (100 MHz, d_6 -DMSO) δ 166.2, 152.4, 139.6, 136.4, 135.9, 135.7, 130.2, 129.2, 128.7, 128.7, 128.5, 128.0, 127.3, 126.8, 125.3, 124.1, 82.8, 72.7, 53.3, 42.6, 27.8.

6. There are some up-to-date literatures documented the multicomponent polymerizations involving alkyne and sulfonyl azide monomers. The related literature should be thoroughly checked and cited.

Our response:

We have checked the up-to-date literatures documented the multicomponent polymerizations involving alkyne and sulfonyl azide monomers, and cited in references 19-21 as follows:

19. Lee, I.-H., Bang, K.-T., Yang, H.-S. & Choi, T.-L. Recent Advances in Diversity-Oriented Polymerization Using Cu-Catalyzed Multicomponent Reactions. *Macromol. Rapid Commun.* 2100642 (2021).

20. Bang, K.-T., Kim, H., Kang, S.-Y., Bhaumik, A., Ahn, S., Yun, N. & Choi, T.-L. Constructing a Library of Doubly Grafted Polymers by a One-Shot Cu-Catalyzed Multicomponent Grafting Strategy. *Macromolecules.* **54**, 5539-5548 (2021).

21. Huang, Y., Xu, L., Hu, R. & Tang, B. Z. Cu(I)-Catalyzed Heterogeneous Multicomponent Polymerizations of Alkynes, Sulfonyl Azides, and NH₄Cl. *Macromolecules.* **53**, 10366-10374 (2020).

7. There are a few typos existed in the manuscript. For example, line 147, “formaton”; I do not think the structure of “Int 7” could be claimed as a “dimer”; no “Scheme 3” found in the manuscript.

Our response:

We have revised “formation” in line 147.

We have delete “dimer” for Int.

We have corrected “Scheme 3” to “Fig. 8”.

REVIEWERS' COMMENTS

Reviewer #1 (Remarks to the Author):

The authors have revised the manuscript according to the suggestion and provided more evidence to support the mechanism. It was suggested to be accepted after minor revision.

The assignment of peaks of HRMS spectra in page S7 seems incorrect. The calculated exact mass of the structures is 247.1109 and 219.1048. Thus the m/z signals (240.1266 and 219.1042) is not related to the proposed D and E. It should be checked carefully and confirmed.

Reviewer #2 (Remarks to the Author):

The authors have made considerable efforts to address the questions raised by both two reviewers. Necessary additional experimental evidences were provided in the revised manuscript. I think the manuscript can now be recommended for the publication in Nature Communications.

To Reviewer #1:

The authors have revised the manuscript according to the suggestion and provided more evidence to support the mechanism. It was suggested to be accepted after minor revision. The assignment of peaks of HRMS spectra in page S7 seems incorrect. The calculated exact mass of the structures is 247.1109 and 219.1048. Thus the m/z signals (240.1266 and 219.1042) is not related to the proposed D and E. It should be checked carefully and confirmed.

Our response:

We have examined the HRMS peaks in page S7. The carbene intermediate D was protonated to form more stable structure and the m/z 249.1259 was observed as [M+2H] (calcd is 249.1266 for [M+2H]). The alkylidene ketenimine E could also be found at m/z 219.1042 (calcd is 219.1048). The differences between experiment values and calculated values are within the allowable range of error due to the HRMS spectra instrumental issues (-0.0007 for intermediate D and -0.0006 for intermediate E). In order to avoid any misunderstanding, we also revised the descriptions for intermediate D in page S7.

To Reviewer 2:

The authors have made considerable efforts to address the questions raised by both two reviewers. Necessary additional experimental evidences were provided in the revised manuscript. I think the manuscript can now be recommended for the publication in Nature Communications.

Our response:

We appreciate for Reviewer 2' comments and suggestions.